# Addressing Sample Complexity in Visual Tasks Using HER and Hallucinatory GANs

**Himanshu Sahni**[†][*]    **Toby Buckley**[‡]    **Pieter Abbeel**[‡, §]    **Ilya Kuzovkin**[‡]

[†] Georgia Institute of Technology
[‡] OffWorld Inc.
[§] University of California, Berkeley

## Abstract

Reinforcement Learning (RL) algorithms typically require millions of environment interactions to learn successful policies in sparse reward settings. Hindsight Experience Replay (HER) was introduced as a technique to increase sample efficiency by reimagining unsuccessful trajectories as successful ones by altering the originally intended goals. However, it cannot be directly applied to visual environments where goal states are often characterized by the presence of distinct visual features. In this work, we show how visual trajectories can be hallucinated to appear successful by altering agent observations using a generative model trained on relatively few snapshots of the goal. We then use this model in combination with HER to train RL agents in visual settings. We validate our approach on 3D navigation tasks and a simulated robotics application and show marked improvement over baselines derived from previous work.

## 1   Introduction

Deep Reinforcement Learning (RL) has recently demonstrated success in a range of previously unsolved tasks, from playing Atari and Go on a superhuman level [23, 14, 34] to learning control policies for real robotics tasks [20, 28, 29]. But deep RL algorithms are highly sample inefficient for complex tasks and learning from sparse rewards can be challenging. In these settings, millions of steps are wasted exploring trajectories that yield no learning signal. On the other hand, shaping the rewards in an attempt to make learning easier is non-trivial and can often lead to unexpected 'hacking' behaviour [26, 30]. Therefore, an important vector for RL research is towards more sample efficient methods that minimize the number of environment interactions, yet can be trained using only sparse rewards.

To this end, Andrychowicz et al. [1] introduced Hindsight Experience Replay (HER), which can rapidly train goal-conditioned policies by retroactively imagining failed trajectories as successful ones. HER was able to learn a range of robotics tasks that traditional RL approaches are unable to solve. But it was only shown to work in non-visual environments, where the state input is composed of object locations and proprioceptive features and it is straightforward to convert any state into a goal. The precise goal configuration is provided to the agent's policy throughout training through a universal value function approximator (UVFA) [31]. UVFAs provide a simple mechanism for reimagining goals by allowing direct substitution of a new goal in off-policy settings. In many visual environments, though, goal states appear different from other states. Moreover, if the agent's policy is conditioned solely on its state, goals states have to be sought out in the state image using their distinct visual cues and, in order to reimagine goals, the agent's observations themselves must be

---

[*]Work done as an intern at OffWorld Inc. Correspondence to: hsahni3@gatech.edu

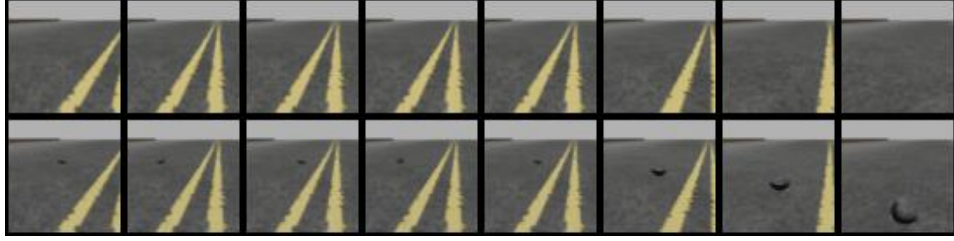

Figure 1: HALGAN hallucinates the presence of goals in unsuccessful trajectories, ending in a perceived success. In this environment, the agent's task is to search for a pebble randomly placed in its surroundings and collect it by approaching and centering it in its view. The top row shows a failed trajectory during exploration. The bottom row replays the same trajectory with a hallucination inserted by HALGAN at each step such that a pebble appears to be collected in the final state.

altered retroactively. HER is not directly applicable to such tasks as it provides no such mechanism for altering agent observations.

Yet, we desire for RL agents to quickly learn to operate in the complex visual environments that humans inhabit. To make progress towards this, we introduce a new algorithm for visual hindsight experience replay that addresses the high sample complexity of RL in such visual environments by combining a hallucinatory generative model, HALGAN, with HER to rapidly solve tasks using only state images as input to the agent policy. To retroactively hallucinate success in a visual environment, the failed trajectory of state observations must be altered to appear as if the goal was present in its new location throughout. HALGAN minimally alters images in snippets of failed trajectories to appear as if the desired goal is achieved by the end (see figure 1). HALGAN is trained using relatively few snapshots of *near goal* images, where the relative location of the agent to the goal is annotated beforehand. It is then combined with HER during reinforcement learning, where the goal location is unknown but agent location can be estimated, to hallucinate goals in desired locations along unsuccessful trajectories in hindsight. We primarily focus on tasks where the completion of a goal can be visually identified within the agent state.

The key contributions of this work are to expand the applicability of HER to visual domains by providing a way to retroactively transform failed visual trajectories into successful ones and hence allow the agent to rapidly generalize across multiple goals using only the state as input to its policy. We aim to do so in conjunction with minimizing the amount of direct goal configuration information required to train HALGAN. We believe that the sample complexity reduction HALGAN provides is an important step towards being able to train RL policies directly in the real world.

## 2   Background

**Reinforcement Learning.** In RL, the agent is tasked with the maximization of some notion of a long term expected reward [35]. The problem is typically modeled as a Markov decision process (MDP) $< S, A, R, T, \gamma >$, where $S$ is the set of states the agent can exist in, $A$ is the set of environment actions, $R : S \times A \to \mathbb{R}$ is a mapping from states and actions to a scalar reward, $T : S \times A \to S$ is the transition function, and $\gamma \in [0, 1)$ is a discount factor that weighs the importance of future rewards versus immediate ones. The goal of learning is often the optimal policy, $\pi^* : S \to A$, mapping every state to an action that maximizes the expected sum of future discounted rewards, $\mathbb{E}[\sum_k \gamma^k R(s_{t+k})]$. This expectation is known as the value function ($V : S \to \mathbb{R}$). UVFAs [31] approximate the value function with respect to a goal in addition to the state, $V : S \times G \to \mathbb{R}$. The optimal policy, $\pi^*(s; g)$, in this case, maximizes the probability of achieving a particular goal, $g$, from any state.

Off-policy RL algorithms can learn an optimal policy using experiences from a *behavior policy* separate from the optimal policy. In particular, off-policy algorithms can make use of samples collected in the past, leading to more sample efficient learning. An experience replay [22] is typically employed to store past transitions as tuples of $(s_t, a_t, r_t, s_{t+1})$. At every step of training, a minibatch of transitions is sampled from the replay at random and a loss on future expected return minimized. The off-policy algorithms employing an experience replay used in this work are Double Deep Q-Networks (DDQN) [36] and Deep Deterministic Policy Gradients (DDPG) [21].

**Hindsight Experience Replay.** The essential idea is to store each trajectory, $Traj_i = s_0^i, s_1^i, ..., s_T^i$, with a number of additional goals, typically future agent states, along with the originally specified goals. An off-policy algorithm employing an experience replay is used in conjunction with a UVFA that allows for direct substitution of new goals in hindsight. The reward is also modified retroactively to reflect the new goal being replayed. In particular, HER assumes that every goal, $g \in G$, can be expressed as a predicate $f_g : S \rightarrow \{0, 1\}$. That is to say, all states can be judged as to whether or not a goal $g$ has been achieved in them. Thus, while replaying a trajectory with a surrogate goal $\overline{g}$, one can easily reassign rewards along the entire trajectory as

$$r_{\overline{g}}(s_t^i) = \begin{cases} 1 & if f_{\overline{g}}(s_t^i) = 1 \\ 0 & otherwise. \end{cases}$$

Andrychowicz et al. [1] report that selecting $\overline{g}$ to be a future state from within the same (failed) episode leads to the best results. This training approach forms a sort of implicit curriculum for the agent. In the beginning, it encourages the agent to explore further outwards along trajectories it has visited before. The agent soon learns to associate the hindsight rewards with the surrogate goals, $\overline{g}$. Over time, the agent is able to generalize to achieve any goal in $G$.

**Wasserstein GANs.** We employ a Wasserstein ACGAN [11, 27] as our generative model because of its stability, realistic outputs, and ability to condition on a desired class. A typical W-ACGAN has a generator, $H$, that takes as input a class variable and a latent vector of random noise. It generates an image that is fed into the discriminator, $D$ which rates the image on fidelity to the training data. As an auxiliary task, $D$ also predicts class membership. The Wasserstein distance between the distributions of real, $p_R$, and generated, $p_H$, images is used as a loss to train the combined model. W-ACGANs produce realistic looking hallucinations that will allow the agent to easily generalize from imagined goal states to real ones. Realistic insertion of goals was not an issue in HER because a new goal could directly be substituted in a replayed transition without any modification to the observations.

## 3 Related Work

**Generative Models in RL.** In recent years, generative models have demonstrated significant improvements in the areas of image generation, data compression, denoising, and latent-space representations, among others [9, 2, 16, 5, 37]. Reinforcement learning has also benefited from incorporating generative models in the training process. Ha and Schmidhuber [12] unify many approaches in the area by proposing a Recurrent Neural Network (RNN) based generative dynamics model [32] of popular OpenAI gym [3] and VizDoom [17] environments. They employ a fairly common procedure of encoding high dimensional visual inputs from the environment into lower dimension embedding vectors by using a Variational Auto Encoder (VAE) [18] before passing it on to the RNN model. Another approach , GoalGAN [13], uses a GAN to generate goals of difficulty that matches an agent's skill on a task. But it assumes that goals can easily be set in the environment by the agent and does not make efficient use of trajectories that failed to achieve these objectives. Generative models have also been used in the closely related field of imitation learning to learn from human demonstrations or observation sequences [15, 8, 33]. Our approach does not require demonstrations of the task, or even a sequence of observations, only relatively few random snapshots of the goal with a known configuration which we use to speed up reinforcement learning.

**Goal Based RL.** Some recent work has focused on leveraging information on the goal or surrounding states to speed up reinforcement learning. Edwards et al. [7] and Goyal et al. [10] learn a reverse dynamics model to generate states backwards from the goal which are then added to the agent's replay buffer. The former work assumes that the goal configuration is known and backtracks from there, whereas in the latter, high-value states are picked from the replay buffer or a GoalGAN is used to generate goals. The latter work also learns an inverse policy, $\pi(a_t|s_{t+1})$ to generate plausible actions leading back from goal states. In contrast, we focus on minimally altering states in existing failed trajectories already in the replay buffer to appear as if a goal has been completed in them. This avoids having to generate entirely new trajectories and allows us to make full use of the environment dynamics already present in previous state transitions.

Others have focused on learning goal-conditioned policies in visual domains by using a single or few images of the goal [39, 41]. Nair et al. [25] train a $\beta$-VAE [4] on state images for a threefold purpose: (1) to sample new goals during training, (2) to use the Euclidean distance between feature encodings of current and goal images as a dense reward, and (3) to retroactively substituted goals with images

generated by the VAE and reassign rewards appropriately. Here also, the set of goals $G$ is assumed to be the same as the set of states $S$, i.e. goal states appear similar to regular states and hence they are easy to swap back and forth. This works well for domains where the goal is separately provided to the policy along with the agent state, and where states do not have to be modified for changing goals. In this work, we attempt learning in domains where the where the goal may or may not be present in a particular agent state and hence has to be added in during hindsight and the goal image is not separately provided to the agent's policy.

## 4  The missing component in HER

First, we will formally discuss what is missing from the original HER formulation that does not allow it to readily extend to visual domains. In the next section, we describe in detail how the use of hallucinatory generative models can help bridge the gap.

Andrychowicz et al. [1] make the assumption that "given a state $s$ we can easily find a goal $g$ which is satisfied in this state". This requires a mapping, $m : S \rightarrow G$ from every state $s \in S$ to a goal $g \in G$ that is achieved by the agent being present in $s$. While this mapping may be relatively straightforward to hand design for real-valued state spaces, its analog for visual states cannot be constructed easily. For example, if $S$ is the plane of real values in $\mathbb{R}^2$, the goal may be to achieve a particular $x$-coordinate. So in the state $(x = 0.5, \ y = 1.0)$, a goal that is satisfied is simply $\overline{g} : x = 0.5$. But in visual environments, *goal* states may have visual features distinct from regular states. Imagine if the agent must instead navigate to a beacon on a 2D plane using camera images as input. In order to convert a state into one in which a goal is satisfied, the beacon must be visually inserted into the state image itself. In this case, a function capable to mapping states to goals is difficult to hand design.

In order to fully utilize the power of HER, not only should the agent be able to hallucinate goals in arbitrary states, but also consistently in the same absolute position throughout the failed trajectory. Note that with each step along the trajectory, the position of the goal (the beacon) changes relative to the agent's and thus the agent's observation must be correctly updated to reflect this change. The goal must *appear* to be solved in a future state along every step of the trajectory in a way that is consistent with the environment dynamics. Only then can we make use of the existing transitions along the trajectory for replay with hallucinated as well as original goals. Thus, visual settings require the mapping $m$ to be extended along the entire trajectory, $m_V : S_{Traj}^T \rightarrow G$, where $Traj$ is the space of failed trajectories and $T$ is the maximum length of a trajectory snippet. Every state along the trajectory, $s_0, s_1, \ldots, s_T \in S_{Traj}$, must be modified by the mapping into a *near goal* state, $\overline{s_0}, \overline{s_1}, \ldots, \overline{s_T}$, that is consistent with the final hallucinated goal state, $\overline{s_T} = \overline{g}$ (see figure 1). This work's main contribution lies in showing that such a mapping can be learned by a generative model using some knowledge of the goal in the form of *goal snapshots* with known relative location.

Lastly, the use of UVFAs does not extend to visual settings where the agent's policy is not conditioned on a specific goal location, but where a desired goal must be searched for within the environment using visual cues, such as navigating to a beacon. We show how the learned model mapping unsuccessful trajectories to successful ones can be applied to rapidly train RL agents with policies solely conditioned on their state image.

## 5  Approach

To address the shortcomings of HER in visual domains, we adopt a two-part approach. First, a generative model, HALGAN, is trained to modify any existing state from a failed trajectory into a goal or *near goal* state. Then, during reinforcement learning, HALGAN generates goal hallucinations conditioned on the configuration of the agent in the current state relative to its own configuration in a future state from the same episode. Details on each component of HALGAN and how it all fits together to generate consistent hallucinations of the goal are discussed next.

### 5.1  Minimal Hallucinations of Visual Goals

Our aim is to minimally alter a failed trajectory in order to turn its states into goal or *near-goal* states. This makes full use of existing trajectories and does not require HALGAN to re-imagine the environment dynamics or unnecessary details about the goal state such as the background.

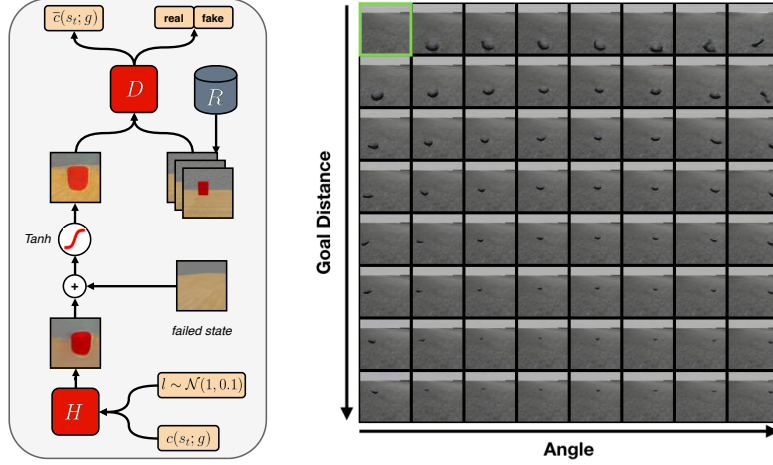

Figure 2: (left) A schematic of the HALGAN training process. $c(s_t; g)$ informs the generator, $H$, on the desired location of the goal. The generated image is added to a random state from the replay, renormalized and passed to the discriminator. $D$ rates its authenticity and predicts goal location. (right) HALGAN output for different relative positions of the goal. The original failed state is on the top left. Demonstrates the fidelity to real states and ability to accurately control goal placement.

To this end, we train an additive model such that the generator, $H$, has to produce only differences to the state image that add in the goal. To obtain a hallucinated image $\overline{s_t}$ with the goal at the final state of the trajectory, $s_T$, we compute,

$$\overline{s_t} = Tanh\left(s_t + H\left(c(s_t; s_T), l\right)\right), \tag{1}$$

where $c(s_t; g)$ is the relative configuration of the robot to a desired goal state $g$ and $l$ is a random latent conditioning vector. $Tanh$ re-normalizes the hallucinated state image to $[-1, 1]$. Note that hallucinations are generated independent of other states in the trajectory. Temporal consistency between hallucinations on two consecutive states of a failed trajectory is only enforced through the relative configuration to a common final desired goal state.

The hallucinated state, $\overline{s_t}$, along with a state $s_r$ sampled from dataset $R$, is then fed to the discriminator $D$ to compute the discriminative loss,

$$L_D = \mathbb{E}_{\overline{s_t} \sim p_H}[log D(\overline{s_t})] - \mathbb{E}_{s_r \sim p_R}[log(D(s_r))]. \tag{2}$$

As a result of generating only image differences, the trained hallucinatory model is invariant to some kinds of visual variations, such as background, presence of other objects, etc.

Additionaly, a gradient penalty is typically employed in the training of Wasserstein GANs [11].

$$L_\nabla = \mathbb{E}_{\hat{s} \sim P_{\hat{s}}}\left(\|\nabla D(\hat{s})\|_2 - 1\right)^2 \tag{3}$$

To further encourage the model to generate minimal modifications to the original failed image, we also add a $L_2$ norm loss on the output of $H$. In our experiments, this helped remove unnecessary elements in the hallucinations such as multiple goals or background elements such as walls.

$$L_H = \|H\left(c(s_t; s_T), l\right)\|_2 \tag{4}$$

## 5.2 Regression Auxiliary Task

Typical ACGANs are conditioned on a discrete set of classes, such as flower, dog, etc [27]. In our approach, the generator is conditioned on the relative configuration of the agent from the desired goal state, which is a real-valued vector $c(s_t; g) \in \mathbb{R}^n$. The auxiliary task for the discriminator is to regress to the real valued relative location of the goal seen in a training image. To train this regression based auxiliary task, we use a *mean squared error* loss,

$$L_A = \|\overline{c}(\overline{s_t}) - c(s_t; g)\|_2 \tag{5}$$

where $\overline{c}(\overline{s_t})$ is the relative configuration predicted by $D$. We found it helpful to add a small amount of Gaussian noise to the auxiliary inputs for robust training, especially on smaller datasets.

## 5.3 HALGAN

Our final loss to the combined HALGAN is,

$$L = L_D + \alpha L_\nabla + \beta L_H + \lambda L_A \tag{6}$$

where, $\alpha$, $\beta$, and $\lambda$ are weighting hyperparameters, which we fix to 10, 1, and 10 respectively.

To summarize, the training procedure is as follows. $H$, conditioned on a randomly drawn desired relative goal location produces a hallucination which is then added to a randomly selected image from a failed trajectory. The discriminator is provided these hallucinated images, as well as ground truth images from $R$ and has to score them on their authenticity and also predict the relative goal location. See figure 2 (left) for a representation of the HALGAN training process and the appendix for more details on the network architectures and training hyperparameters. Figure 2 (right) shows examples of the output from our model for a range of goal configurations.

**Data Collection.** HALGAN is trained on a dataset, $R$, of observations of the goal where the relative configuration to the agent is known. These snapshots of the goal can be collected and annotated before RL and are only used once to train HALGAN. During RL, hallucinations are created using only the agent's own configuration, which can be obtained in realistic applications using SLAM or other state tracking techniques [24]. For the purposes of our experiments, we collect the training data in $R$ by using the last 16 or 32 states of a successful rollout. At the end of a successful rollout, assuming that the agent's configuration corresponds to the goal location relative poses can be calculated automatically using only agent configuration. We did not manually inspect all images in $R$ to ensure that the goal is visible, but there was enough relevant data for HALGAN to infer the object of interest. Note that the exact data required are randomly selected snapshots from near the goal, in any order. Only observations of the goal along with the annotated relative configurations are used, no actions have to be provided or demonstrated, which allows the generative model to be independent of the agent and demonstrator action spaces. Thus, the burden of collecting goal information for HER is not entirely eliminated, but can be significantly reduced to a few thousand states. We also collect a dataset of failed trajectories. Most off-policy RL methods that employ an experience replay have a *replay warmup* period where actions are taken randomly to fill the replay to a minimum before training begins. This dataset of failed trajectories can be the same as the replay warmup.

## 5.4 Visual HER Using HALGAN

During reinforcement learning, the agent explores the environment using its behaviour policy. Snippets of past trajectories are sampled from the experience replay at every step and a few of the failed ones are augmented with goal hallucinations to appear successful. Again, this is in contrast to the regular HER approach or the approach by Nair et al. [25], where end states were directly designated as goals using a hand-designed mapping and the observations in the failed trajectory did not have to be modified. The detailed process is explained in algorithm 1. The result is that the agent encounters hallucinated *near goal* states with a much higher frequency than if it were randomly exploring. This in turn encourages the agent to explore close to real *near goal* states.

An important consideration is the retroactive reassignment of rewards. HER uses a manually defined function $f_g(s)$, which decides if the goal $g$ is satisfied in state $s$, to designate rewards during hindsight. This sort of reward function is hard to hand design in visual environments. Comparing state and goal images pixel by pixel is typically ineffective. For the purpose of hindsight replay where a future state is set as the goal, one needs only to compare two states to reassign rewards, $f_s : S \times S \rightarrow \{0, 1\}$. As mentioned in section 3, Nair et al. [25] use a trained $\beta$-VAE as $f_s$ to reassign rewards in a dense manner. Here, we make use of access to the agent's own configuration. We assume that any two states with similar configuration must satisfy the same goal. During retroactive reward reassignment, we compare the configuration of the agent in the sampled state to that at the end of the trajectory. A sparse reward of +1 is awarded if they are the same up to a threshold value.

**Algorithm 1** HALGAN+HER

---

1: **Given:** Trained hallucinatory model $H$, Reward reassignment strategy $r_g(s)$.
2: Initialize off-policy Algorithm $\mathbb{A}$.             $\triangleright$ eg. DDQN, DDPG
3: Initialize Experience Replay $E$ by random exploration.
4: **for** step$= 1, N$ **do**
5:      Sample an action according to behavior policy $a_t \leftarrow \pi(s_t)$ in current state.
6:      Execute $a_t$ in the environment and observe state $s_{t+1}$, reward $r_t$.
7:      Store tuple $\langle s_t, a_t, r_t, s_{t+1}\rangle$ in $E$.
8:      Sample minibatch $B$ from $E$ for training.
9:      **for** $e = \langle s_i, a_i, r_i, s_{i+1}\rangle$ in $B$ **do**
10:         **if** $c \sim Bern(p)$ **then**             $\triangleright$ $p =$ hallucination prob.
11:             Sample $d \sim Unif(\{0, 1, ..., D\})$        $\triangleright$ distance to goal state
12:             Compute relative configurations $c(s_i; s_{i+d})$ and $c(s_{i+1}; s_{i+d})$.
13:             $s_i \leftarrow s_i + H(c(s_i; s_{i+d}), l)$
14:             $s_{i+1} \leftarrow s_{i+1} + H(c(s_{i+1}; s_{i+d}), l)$
15:             $r_i \leftarrow r_{s_{i+d}}(s_{i+1})$
16:      Perform one step of optimization using $\mathbb{A}$ on the modified minibatch $B$.

---

# 6 Experiments

We test our method on two first person visual environments. In a modified version of MiniWorld [6], we design two tasks. The first one is to *navigate* to a red box located in an enclosed room (figure 3a top). The second task is to *successively navigate*, first to the red box, picking it up by visually centering it, and then carrying it to a green box somewhere else in the room (see figure 3a bottom). The second environment is a more visually realistic simulated robotics domain, where a TurtleBot2 [38] equipped with an RGB camera is simulated within Gazebo [19]. We use gym-gazebo [40] to interface with Gazebo. Here, the agent must collect a pebble scattered randomly on a road by approaching and centering it in its visual field (figure 3b). The environment only provides a sparse reward of 1 for achieving the goal. We demonstrate that our method applies easily to both discrete (TurtleBot) and continuous control (MiniWorld) environments.

The size of the near goal dataset, $R$, for the Turtlebot, *navigation* and *successive navigation* tasks is 6840, 2000, and 6419 images with relative goal configurations respectively but we also show results on smaller datasets for the Turtlebot environment (figure 3f). In the Turtlebot and MiniWorld *navigation* tasks, the configuration of the agent is simply it's $\langle x, y, yaw\rangle$. In *successive navigation*, an additional binary field indicates whether the red box is held by the agent. The agent's relative configuration is calculated with respect to the red box before it is picked up, and the green box afterwards. Hallucinations are generated for the agent approaching both boxes. We found it helpful to anneal the amount of hallucinations in a batch over time as the agent fills its replay with real goal images. Details of all experimental hyperparameters are provided in the appendix.

**Comparisons.** There is no prior work that attempts HER in visual domains without explicit goal conditioning. Hence, we compare our approach to multiple extensions of existing approachs and standard RL baselines. All baselines used the same hyperparameters as our approach. First, a naive extension of HER into the visual domain, *her*, simply rewards the agent for states at the end of failed trajectories during replay without hallucinating. A second baseline is derived from *RIG* [25] which trains goal-conditioned policies with a dense reward based on the distance between the embedding of the sampled state and that of a goal image. *RIG*'s reimagining of goals relies on the use of UVFAs, which is not possible for our domains where the goal image is unknown. Therefore, we design two variants of this baseline in an attempt to find a suitable comparison. For both, we first train a VAE on *near goal* images in $R$ and failed state images. Then, during RL, *vae-her* simply sets the final image in a failed trajectory as the goal and uses the trained VAE to reassign reward for a transition along that trajectory. *rig-* follows a similar dense reward shaping strategy, but computes distance of a state to a randomly sampled goal image in $R$. Hence, *rig-* rewards the agent for being in states that look similar to goal states in $R$. The distance based rewards provided by the VAE in *rig-* had to be re-scaled by a constant factor of $0.02$ to be the same order of magnitude as the environment rewards.

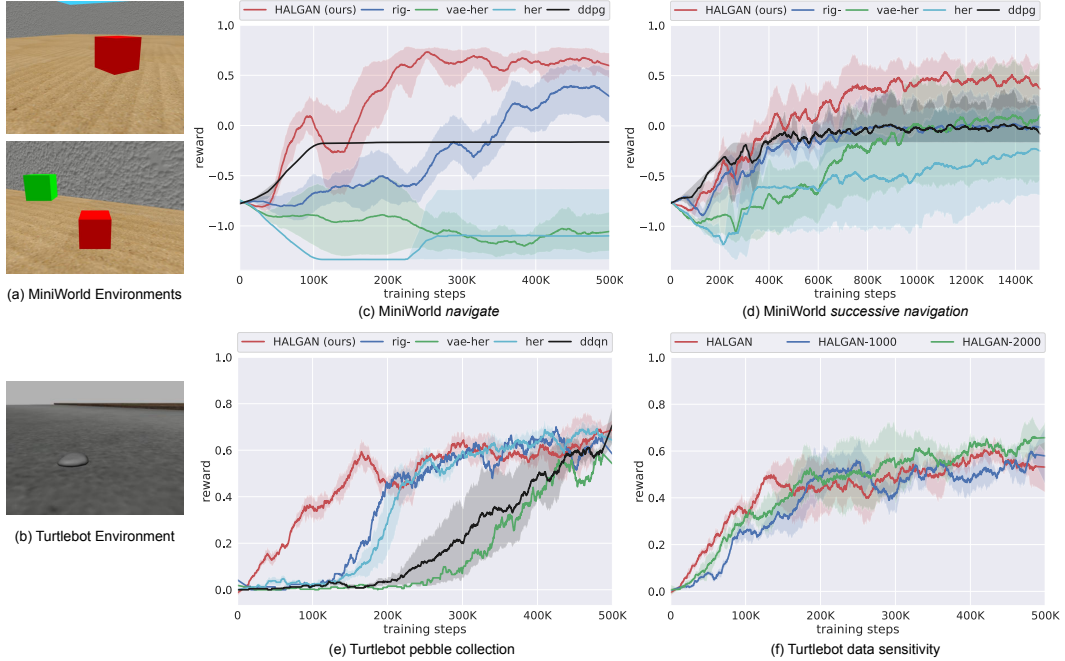

Figure 3: (a,b) Near goal states in experimental domains. (c,d) Episodic reward vs. environment steps for MiniWorld tasks averaged over 5 random seeds. (e) Results for Turtlebot task are averaged for 3 random seeds. (f) Minor variance in agent performance in Turtlebot task with decreasing size of HALGAN training dataset. 90% confidence intervals are shown for each plot.

# 7  Results

In all of our experiments, HALGAN trained agent begins learning immediately (figure 3). This is due to the realistic looking hallucinated goals being quickly identified as desirable states. This incentivizes the agent to explore more around goal states. This is in contrast to standard RL which rarely encounters reward and must explore at length in order to begin the learning process, if at all.

In the discrete TurtleBot pebble collection domain (figure 3e), the naive HER strategy provides a good enough exploration bonus for the agent to explore further and quicker than standard DDQN. It begins learning by 100K steps. The *rig-* baseline performs only slightly better. HALGAN agent, by contrast, starts learning to navigate to real goals immediately.

For the continuous control experiments in MiniWorld (figure 3c, 3d), only HALGAN agent is able to learn to complete the task. Note that only positive rewards indicate achievement of goal. DDPG never encounters any reward during exploration and hence learns to minimize its actions in order to avoid movement penalty. Naive *her* initially encourages exploration and hence incurs a heavy penalty, but does not learn to associate the hallucinated rewards with the presence of a goal. *vae-her*, the augmentation of *her* with dense rewards from a trained VAE, also proves unsuccessful for either task, demonstrating that dense rewards without hallucinated or real goals in failed trajectories are ineffective for learning in these domains. Only the *rig-* strategy of providing dense rewards relative to random goal images eventually learns to complete the *navigation* task for some of the seeds. For the *successive navigation* task, *rig-* only learns a working policy on a single seed and the other baselines perform similarly or worse. Interestingly, *rig's* dense reward reassignment can be readily combined with our approach of modifying observations, providing directions for future work.

Finally in figure 3f, we show the change in performance on the TurtleBot task due to using fewer training samples in $R$. The effect is only slightly slower learning even for the largely reduced dataset of only 1000 images. The minimalistic hallucinations created by HALGAN require a relatively small amount of training data to provide a significant boost in reinforcement learning.

## 8 Discussion

A major impediment to training RL agents in the real world is the amount of data an agent must collect before it encounters rewards and associates them with goals. High sample complexity makes problems such as fragility of physical systems, energy consumption, speed of robots and sensor errors manifest themselves acutely. In this work, we have shown that Hindsight Experience Replay can be extended to visual scenarios by retroactively hallucinating goals into agent observations. We empirically demonstrate that by utilizing failed trajectories in such a way, the agent can begin learning to solve tasks immediately. HALGAN+HER trained agent converges faster than standard RL techniques and derived baselines on navigation tasks in a 3D environment and on a simulated robot.

The principle of visually hallucinating goals could potentially be applied to speed up training for many other tasks, for example, avoiding collisions (negative penalty on hallucinated collisions), following a human or object (positive reward for hallucinations within a range of distance), or placing objects in a visually identified zone (hallucinating a visual safety marker). HALGAN is currently conditioned solely on relative agent configuration. Complex visual environments may include cues in the background that influence goal appearance, such as occlusions or lighting. Conditioning HALGAN on features of the current or intended goal state could extend this approach to such environments. Conditioning on a history of states in the trajectory could also enforce further temporal consistency between hallucinations. Other future directions of work include collecting training data for HALGAN online as the agent explores and automatically annotating goal configuration.

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
