[Supplementary Material]

# Appendix

## A. Experimental Hyperparameters

Refer to table below for environment specific hyperparameters.

| HYPERPARAMETER | TURTLEBOT | MINIWORLD NAVIGATE | MINIWORLD PICK-AND-PLACE |
|---|---|---|---|
| REPLAY WARMUP | 10,000 | 10,000 | 10,000 |
| REPLAY CAPACITY | 100,000 | 100,000 | 100,000 |
| INITIAL EXPLORATION $\epsilon$ | 1.0 | 1.0 | 1.0 |
| FINAL EXPLORATION $\epsilon$ | 0.5 | 0.5 | 0.5 |
| $\epsilon$ ANNEAL STEPS | 100,000 | 100,000 | 250,000 |
| DISCOUNT ($\gamma$) | 0.99 | 0.99 | 0.99 |
| OFF-POLICY ALGORITHM | DDQN | DDPG | DDPG |
| POLICY OPTIMIZER | ADAM | ADAM | ADAM |
| LEARNING RATE | $1e^{-3}$ | $1e^{-5}$ (ACTOR), $1e^{-4}$ (CRITIC) | $1e^{-5}$ (ACTOR), $1e^{-4}$ (CRITIC) |
| SIZE OF $R$ FOR HALGAN | 6,840 | 2,000 | 6,419 |
| HALLUCINATION START % | 20% | 30% | 30% |
| HALLUCINATION END % | 0% | 0% | 0% |
| MAX FAILED TRAJECTORY LENGTH | 16 | 32 | 16 |
| IMAGE SIZE | 64x64 | 64x64 | 64x64 |
| RANDOM SEEDS | 75839, 69045, 47040 | 75839, 69045, 47040, 60489, 11798 | 75839, 69045, 47040, 60489, 11798 |

*Table 1.* Environment Specific Hyperparameters

Refer to table below for HALGAN specific hyperparameters.

| HYPERPARAMETER | VALUE |
|---|---|
| LATENT VECTOR SIZE | 128 |
| LATENT SAMPLING DISTRIBUTION | $\mathcal{N}(1, 0.1)$ |
| AUXILIARY TASK WEIGHT | 10 |
| GRADIENT PENALTY WEIGHT | 10 |
| $L_2$ LOSS ON $H$ WEIGHT | 1 |
| OPTIMIZER | ADAM |
| LEARNING RATE | $1e-4$ |
| ADAM $\beta1$ | 0.5 |
| ADAM $\beta2$ | 0.9 |
| $D$ ITERS PER $H$ ITER | 5 |

*Table 2.* Hyperparameters involved in training HALGAN

# B. Network Architectures

Refer to table below for details on the network architecture for DDQN. LeakyReLu's were used as activations throughout except for the output layer where no activation was used.

| LAYER | SHAPE | FILTERS | #PARAMS |
|---|---|---|---|
| IMAGE INPUT | 64x64 | 3 | 0 |
| CONV 1 | 5x5 | 4 | 304 |
| CONV 2 | 5x5 | 8 | 808 |
| CONV 3 | 5x5 | 16 | 3216 |
| CONV 4 | 5x5 | 32 | 12832 |
| DENSE 1 | 32 | - | 16416 |
| DENSE 2 | 4 (*nbactions*) | - | 132 |
| TOTAL | - | - | 33708 |

*Table 3.* Network Architecture for DDQN Agent

Refer to table below for details on the network architecture for actor for DDPG. LeakyReLu's were used as activations throughout except for the output layer where a Tanh was used.

| LAYER | SHAPE | FILTERS | #PARAMS |
|---|---|---|---|
| IMAGE INPUT | 64x64 | 3 | 0 |
| CONV 1 | 5x5 | 4 | 304 |
| CONV 2 | 5x5 | 8 | 808 |
| CONV 3 | 5x5 | 16 | 3216 |
| CONV 4 | 5x5 | 32 | 12832 |
| DENSE 1 | 32 | - | 16416 |
| DENSE 2 | 2 (*nbactions*) | - | 66 |
| TOTAL | - | - | 33642 |

*Table 4.* Network Architecture for DDPG Actor

Refer to table below for details on the network architecture for critic for DDPG. LeakyReLu's were used as activations throughout except for the output layer where no activation was used.

| LAYER | SHAPE | FILTERS | #PARAMS |
|---|---|---|---|
| IMAGE INPUT | 64x64 | 3 | 0 |
| CONV 1 | 5x5 | 4 | 304 |
| CONV 2 | 5x5 | 8 | 808 |
| CONV 3 | 5x5 | 16 | 3216 |
| CONV 4 | 5x5 | 32 | 12832 |
| DENSE 1 | 32 | - | 16416 |
| DENSE 2 | 1 | - | 33 |
| TOTAL | - | - | 33673 |

*Table 5.* Network Architecture for DDPG Critic

Refer to table below for details on the network architecture for the generator in HALGAN. LeakyReLu's were used as

activations throughout except immediately after the conditioning layer where no activation was used and the output where tanh was used.

| LAYER | SHAPE | FILTERS | #PARAMS |
|---|---|---|---|
| CONFIG INPUT | 3 | - | 0 |
| DENSE 1 | 128 | - | 384 |
| CONDITIONING INPUT | 128 | - | 0 |
| MULTIPLY | 128 | - | 0 |
| RESHAPE | 1x1 | 128 | 0 |
| UPSAMPLE + CONV 1 | 4x4 | 64 | 131136 |
| BATCHNORM | 2x2 | 64 | 256 |
| UPSAMPLE + CONV 2 | 4x4 | 64 | 65600 |
| BATCHNORM | 4x4 | 64 | 256 |
| UPSAMPLE + CONV 3 | 4x4 | 64 | 65600 |
| BATCHNORM | 8x8 | 64 | 256 |
| UPSAMPLE + CONV 4 | 4x4 | 32 | 32800 |
| BATCHNORM | 16x16 | 32 | 256 |
| UPSAMPLE + CONV 5 | 4x4 | 32 | 16416 |
| BATCHNORM | 32x32 | 32 | 128 |
| UPSAMPLE + CONV 6 | 4x4 | 16 | 8028 |
| BATCHNORM | 64x64 | 16 | 64 |
| CONV 7 | 4x4 | 8 | 2056 |
| BATCHNORM | 64x64 | 8 | 32 |
| CONV 8 | 4x4 | 3 | 387 |
| TOTAL | - | - | 323707 |

*Table 6.* Network Architecture HALGAN Generator

Refer to table below for details on the network architecture for the discriminator in HALGAN. LeakyReLu's were used as activations throughout except at the output where no activation was used.

| LAYER | SHAPE | FILTERS | #PARAMS |
|---|---|---|---|
| IMAGE INPUT | 64x64 | 3 | 0 |
| CONV 1 | 4x4 | 32 | 1568 |
| CONV 2 | 4x4 | 32 | 16416 |
| CONV 3 | 4x4 | 32 | 16416 |
| CONV 4 | 4x4 | 64 | 32832 |
| CONV 5 | 4x4 | 64 | 65600 |
| CONV 6 | 4x4 | 64 | 65600 |
| CONV 7 | 4x4 | 128 | 131200 |
| DENSE (AUX) | 2 | - | 129 |
| DENSE (REAL/FAKE) | 1: | - | 258 |
| TOTAL | - | - | 330019 |

*Table 7.* Network Architecture for HALGAN Discriminator