[Reviews · NeurIPS 2019]

Reviewer 1



Originality: There is no prior work that attempts HER in visual domains without explicit goal conditioning. This work is the first one to do this. The designing and training on HALGAN seem rather standard and similar to InfoGAN. Quality: This paper is technically sound and the performance of HALGAN + HER is validated in two environments. Clarity: This paper is well written and easy to follow. The details are adequate for readers to understand and reproduce the paper. Significance: While the authors hope this method could be a step to reducing the sample complexity of training RL and eventually training agents in the real world. The environments shown in the paper are primitive. It is easy to synthesize images in these two environments while it is not clear how HALGAN will perform in diverse and realistic 3D indoor scenes.

Reviewer 2



Originality: The authors make clear the distinction from related work. They are not the first to integrate GANs for generating goals in RL, but do so in a new and interesting way. Quality: The comparison with baselines is thorough, showing the benefit of this approach for these domains. However, page 7 claims that HALGAN RL agents need fewer samples than standard RL, and yet in fact HALGAN must be exposed to enough samples of successful trajectories to be able to effectively hallucinate goal states. Are the 1000-samples used to train the HALGAN shown in Figure 3(f) 1000 examples of /goal/ states, or just states in general. How long does the agent have to explore before a HALGAN can be trained? This discussion needs to be made more clearly and carefully. A second question is why the generator of HALGAN does not input the s_t that it is trying to modify. Without seeing the original state, how can HALGAN generate a goal if the goal is potentially occluded by something in the state. If this can never happen, are these environments realistic? Will the HALGAN approach work in more complex environments? Clarity: Overall the paper is very clear, the diagram in Figure 2 and the rest of the figures contribute to ease of understanding the contributions. The distinction from prior work is clear, as is the motivation. Minor point: there is a missing period at the end of the paragraph which starts section 5 (ending with "hallucinations of the goal are discussed next"). Significance: The idea makes sense and could be useful, however it's not clear how often real-world tasks have visible goals which must be added to the state. Further discussion of examples where this is the case in the real world could help bolster the significance of the paper.

Reviewer 3



The paper extends HER to visual environments where unsuccessful trajectories can be hallucinated to appear successful by altering the agent’s observations using a generative model. While the idea presented in the paper is interesting, I have questions about the scalability of this approach. Specifically, the approach requires training a generative network to produced observations that contain the goal from the environment. I am interested to know how the authors intend to scale up this approach to more complex visual domains like Atari, DeepMind Lab etc. In these complex visual domains hallucinating a goal image that lies within the space of observations of an environment seems difficult and is unclear from the approach presented in this paper. After reading the rebuttal: The main strength and weakness of the paper are as follows (from my perspective): * (strength) the authors introduce a generative approach for applying Hindsight Experience Replay (HER) in visual domains: the idea is simple and has the potential to improve our current Deep RL methods. * (weakness) currently, the paper does not seem to have a detailed discussion on how their generative model was trained to produce images containing the goal information. The authors do clarify this on their feedback and it would be useful if they also add this discussion on their next version of the paper. More importantly, including this discussion is useful for the Deep RL community. * (weakness) their current approach of training the generative model relies on manually annotating the goal images, which may prevent scalability of the algorithm. Addressing this could make their approach be more impactful.

[Author Response · NeurIPS 2019]

We thank the reviewers for their thoughtful feedback. We appreciate the positive comments describing the paper as "new and interesting", "technically sound", "effectively supports claims", "well written" and "easy to follow".

A common question among reviewers was how the approach can be extended to handle more complex visual environments. Indeed this is an important discussion point and a fruitful area for future research. We will integrate the following discussion into the paper.

As R1 points out, this work is the first to extend HER to visual domains without explicit goal conditioning. We show that for certain tasks, retroactive goal reassignment can be done by directly inserting a hallucinated goal into state trajectories while largely leaving the background unaltered. This allows HALGAN to focus on goal generation and not unnecessary background details, making its training sample efficient. Yet, in other visual environments, extra information in the state such as occlusion or background may influence goal generation. For this, the generative model in HALGAN can be extended to condition on extra variables such as the agent state(s) or a map of the environment if available. More broadly, one can think of future methods that use generative models to retroactively alter goals in visual states as lying on a range of how much of the original trajectory they alter and what state variables they condition on. It is also important to note that the advantage of hindsight methods is greatest when a large part of the original failed trajectory can be reused with a new goal.

Now we address individual reviewer questions.

**R1** *"How does it handle more complicated visual input..."* HALGAN synthesizes goal images conditioned solely on desired relative location, hence is independent of clutter or distracting information that may occur in the state. Multiple possible solutions of goal image can be generated by the GAN if there is enough support in the training set. If state dependent goals are desired, the generative model will have to be conditioned on agent state (see above discussion).

*"How does it enforce temporal consistency?"* Temporal consistency is not explicitly enforced by HALGAN. Instead the model generates the hallucinated states in a transition independently, relying on the relative configuration to the final agent state. A change in relative configuration between two states will manifest in a different hallucination by the trained model, but there's no guarantee that goal features independent of relative configuration will be temporally consistent. In practice, for our environments, we found that this was not a burden to RL training as only two consecutive states are used for making off-policy updates. An extension of our approach could maintain some recurrent, hidden state throughout the failed trajectory that is provided as input to a generative model during hallucinations.

**R2** *"Are the 1000-samples used to train the HALGAN shown in Figure 3(f)..."* We include some details in section 5.3 Data Collection, but will expand on them further here and in the paper. The states in figure 3(f) are near goal snapshots, in which the relative configuration of the goal is known. In our experiments, we rely on a pre-collected dataset of the last 16 or 32 states of successful demonstrations, so that relative configurations can be automatically calculated using only the agent state as long as the agent ended at the goal. Alternatively, these data may be collected online as the agent explores by manually annotating a small set of failed initial episodes with relative goal location. Once HALGAN has sufficient training data, it can generalize to future episodes without annotation. The burden of collecting goal information for HER is not entirely eliminated, but can be significantly reduced to only a few thousand states. We did not enforce that the goal be visible in all collected states, but despite this there were enough data for the GAN to infer the object of interest.

*"...why the generator of HALGAN does not input the s_t that it is trying to modify."* Please refer to common discussion above (L6-L15). For tasks where occlusion may play a large role, the generative model can be extended to condition on agent state or a short term memory over the trajectory.

*"examples where this is the case in the real world could help..."* The principle of visually hallucinating goals can be applicable to many other tasks such as avoiding collisions with objects (eg. negative penalty for hallucinations involving hitting pedestrians), following a human/object (eg. positive reward for hallucinations with person at constant distance), placing objects in visually identified zone (eg. hallucinating a visual marker where objects can be safely placed), etc.

**R3** *"...how the authors intend to scale up this approach to more complex visual domains like Atari, DeepMind Lab etc."* With enough training data, HALGAN should work in DeepMind Lab maze tasks of seeking out "apples", which can be hallucinated on the background in the same way as we have shown in our environments. Atari games are generally not amenable to the hindsight family of algorithms as they do not have multiple (visual) goals that can be substituted in retrospect for each other. As such, we have not seen any examples in the literature that attempt HER in Atari or similar environments that do not possess this crucial property. On how to scale to other visual environments where goals may be dependent on state or occlusions may occur, please refer to common discussion above (L6-L15).

We once again thank all reviewers for their useful comments. We will include the responses in the final submission.

[Meta-Review · NeurIPS 2019]

This paper received good scores overall, with little disparity in the final scores. It extends HER to work with visual observations using a GAN to generate samples. The contribution relative to HER is clear. The main weaknesses of this work are lack of details about the generative model training, scalability concerns raised by reviewers and reliance on human annotated frames containing the goal.